# Cognition and Mental Health in Pediatric Patients Following COVID-19

**DOI:** 10.3390/ijerph20065061

**Published:** 2023-03-13

**Authors:** Hadar Avittan, Dmitrijs Kustovs

**Affiliations:** 1Faculty of Medicine, Riga Stradiņš University, Dzirciema Street 16, LV 1007 Riga, Latvia; 2Department of pharmacology, Riga Stradins University, Dzirciema Street 16, LV 1007 Riga, Latvia; dmitrijs.kustovs@rsu.lv

**Keywords:** long COVID, children, cognitive symptoms, delirium, insomnia, mental health

## Abstract

The global coronavirus pandemic has significantly impacted public health and has been a research subject since its emergence in 2019. The acute phase of the disease leads to pulmonary and non-pulmonary manifestations, which in some individuals may progress to long-lasting symptoms. In this article, we conducted a narrative review of the current literature to summarize current knowledge regarding long COVID syndrome in children, focusing on cognitive symptoms. The review included a search of three databases (PubMed, Embase, and Web of Science) using the key phrases “post COVID-19 cognitive pediatric”, “long COVID pediatric”, “mental health long COVID children”, and “COVID-19 cognitive symptoms”. A total of 102 studies were included. The review revealed that the main long-term cognitive symptoms following COVID-19 were memory and concentration deficits, sleep disturbances, and psychiatric states such as anxiety and stress. In addition to the direct physiological effects of a viral infection, there are psychological, behavioral, and social factors contributing to cognitive impairment, which should be addressed regarding the pediatric population. The high prevalence of neurocognitive symptoms in children following COVID-19 emphasizes the importance of understanding the mechanisms of nervous system involvement.

## 1. Introduction

The outbreak of COVID-19 is associated with numerous adverse impacts on societies worldwide. Over 600 million confirmed cases of COVID-19 have been reported by the World Health Organization [1]. According to the American Academy of Pediatrics and Children’s Hospital Association, over 15 million children have been reported to have tested positive for COVID-19 since the onset of the pandemic [2]. Lately, it was suggested that following coronavirus infection, a chronic condition may appear named ‘post-COVID-19’. According to the 2022 Delphi definition developed by Soriano et al., ‘post-COVID-19’ refers to symptoms that often persist three months from the start of the infection with severe acute respiratory syndrome coronavirus 2 (SARS-CoV-2) with symptoms duration of at least two months and which cannot be explained by any other diagnosis [3]. The Department of Health and Human Services defines post-COVID-19 or long COVID as persistent symptoms in patients who have been infected with SARS-CoV-2 [4]. Symptoms commonly associated with post-COVID-19 include shortness of breath, cough, cognitive impairment, and fatigue [3]. Currently, there is a lack of consensus regarding the definition of post-COVID-19 in the pediatric population. Moreover, limited research has focused on the chronic cognitive effects of COVID-19 in children. This syndrome, which has been listed in the ICD-10 classification as “post-COVID-19” since September 2020, encompasses all the variations of the names found in the literature, such as “long-COVID”, or “long-haul COVID” [4]. In this paper, the term “long COVID” is used to refer to this condition.

## 2. Materials and Methods

A narrative literature review was conducted by researching online databases, including MEDLINE (via PubMed), Embase, and Web of Science. We prioritized systematic reviews, meta-analyses, randomized controlled trials, observational (cross-sectional and cohort), and case-control studies, both in children and adults, which were peer-reviewed and written in English. Key phrases used for the search included: “long COVID pediatric”, “post COVID-19 cognitive pediatric”, “memory loss”, “attention control”, “concentration deficits”, “delirium”, “insomnia”, and “mental health long COVID children.” Although not a systematic review, the literature was still evaluated by the two authors H.A and D.K. We included all literature related to COVID-19 in pediatrics published between 1 December 2019 and 15 December 2022. This work aims to provide a summary of the literature on cognitive symptoms in the pediatric population, with a specific emphasis on studies that included children and adolescents aged 19 years or younger. The use of the upper age limit of 19 years in pediatric studies is based on the World Health Organization’s definition of adolescence, which encompasses individuals aged 10 to 19 years [5]. Publications in journals and articles where we failed to access the full text (despite contacting the authors) were excluded. A total of 102 studies were included.

## 3. Results

### 3.1. Cognitive Symptoms

Many symptoms that develop during acute COVID-19 may persist as part of the long COVID syndrome. Along with the characteristic symptoms of acute COVID-19, such as respiratory distress and cough, there are additional symptoms related to cognitive deficits [6]. Furthermore, symptoms that arise during acute COVID-19 may persist as part of the long COVID syndrome. During the acute COVID-19 phase, individuals can experience cognitive symptoms such as encephalopathy-like symptoms, delirium, altered levels of consciousness, and loss of various memory domains. This cluster of symptoms during acute infection is referred to as ‘cognitive COVID’ and may result from a combination of factors, including neurotropism of SARS-CoV-2 and sedation during mechanical ventilation [6]. A case series conducted on nine children aged 4 to 18 years used neuropsychological testing to assess cognitive domains that may potentially be affected by COVID-19 [7]. The testing, conducted through in-person or telehealth (audio-video) visits, found that the patients demonstrated impaired performance primarily in the domain of attention rather than other deficits, such as executive functioning or working memory, which have been observed in infected adults.

It is challenging to study and categorize the cognitive symptoms of long COVID syndrome. Munblit et al. comprised a “Core Outcome Set”, which gives a list of outcomes recommended for assessment in all clinical practice studies of long COVID syndrome and for improving studies in comparability and quality, especially as there is no agreement on the definition of chronic phase of COVID-19 [8]. According to the list, the domain of cognitive functioning aims to quantify outcomes of confusion, concentration impairment, and memory impairment. Most studies on adults have utilized screening tests, such as the Montreal Cognitive Assessment or Mini-Mental State Examination, and they often rely on self-reported data and objective assessment [9]. While some studies on the pediatric population have also used these screening tests, most have limitations in terms of accurate assessment [10]. Zimmermann et al. reviewed over 14 studies in 2021 investigating long COVID syndrome. They found that between 2 and 80% of children and adolescents reported concentration difficulties which may indicate additional cognitive impairments such as memory deficits, impaired information processing speed, poor attention control, and delirium as well as psychiatric symptoms in children following SARS-CoV-2 infection [10]. The precise prevalence and risk factors for the above outcomes remain to be clarified by further research.

Roge et al. conducted a study in Latvia with 236 long COVID syndrome pediatric patients aged 1 month to 18 years. They evaluated four main domains: physical health, mental health, social well-being, and psycho-emotional well-being. The evaluation was conducted through interviews and questionnaire completion. Almost 17% of patients reported concentration difficulties and impaired attention, and over 10% experienced memory impairment [11]. The cognitive deficits previously mentioned were referred to by the authors as “cognitive sequelae”, which is a term that encompasses a range of symptoms, including mood changes, irritability, and anxiety or depression. In their study, which compared COVID-19 patients to non-SARS-CoV-2 infected patients, the authors found that those who recovered from COVID-19 experienced more cognitive symptoms [11]. Additionally, long COVID symptoms were found to be more prevalent among female pediatric patients when compared by sex, but more specifically, females seemed to have a higher incidence of cognitive and neurological symptoms [11].

A meta-analysis by Behnood et al. compared persistent symptoms between confirmed SARS-CoV-2 infected patients to recovered individuals that tested negative for SARS-CoV-2 as study controls. The meta-analysis, which included 22 studies, found that among children and those aged 1 to 19 years, the attributable risk for cognitive deficits was 3% [12]. In addition, the same study found that older age and female sex were associated with an increased risk of persistent symptoms. Overall, the duration of the long COVID symptoms had a median of 125 days, while the cognitive symptoms were associated with a shorter duration. The same meta-analysis revealed that persistent symptoms of cognitive difficulties occur 2–8% more frequently among children who had COVID-19 compared to those without infection [12]. In a cohort study of 120 patients under the age of 18 years diagnosed with SARS-CoV-2 conducted in Barcelona, the patients had persistent COVID-19 symptoms for at least 12 weeks after the diagnosis was made [13]. Among the most common symptoms found, cognitive impairment, such as decreased attention, was present in 44%. The persistent symptoms were present for at least 6 months in 36% of the cases. The evaluation of fatigue was conducted using the ‘Pediatric Functional Assessment of Chronic Illness Therapy-Fatigue’, and health status, including neurological evaluation, was completed using the ‘Pediatric Quality of Life instrument’, which showed that quality of life was impaired in 37.5% of the children and adolescents assessed, and psychosocial health was impaired in 60%, with 23% exhibiting emotional and behavioral problems [13].

An observational study aimed at memory impairment in long COVID took place in a designated pediatric clinic for long COVID syndrome. The study included 90 children with a mean age of 12 ± 5 years and assessed them for at least 410 days (median of 112 days) [14]. Nearly 18% of the patients were found to have memory impairment [14]. Research involving a retrospective chart review of 18 children aged 6-16 years found that 22.2% had memory difficulties [15]. While not one of the most common cognitive symptoms in children, overall studies show that the clinical manifestation of long COVID is similar in both adult and child populations [16]. A recent study by Morand et al. found that the [^18^F]-FDG brain biomarker has been associated with cognitive impairment in long COVID [17]. The study included seven children who complained of fatigue and cognitive impairment, such as concentration and memory deficits, and had [^18^F]-FDG brain PET scans. The scans showed hypometabolism in the pons, cerebellum, and bilateral medial lobes, which contain structures such as the amygdala, uncus, and parahippocampal gyrus. The results of the scans were similar to those observed in adults, supporting the idea that long COVID syndrome in children has common functional brain involvement across age groups, leading to similar symptoms of cognitive deterioration [17].

Currently, little is known about the mechanism behind memory loss in long COVID syndrome. Studies in adults have suggested that cognitive symptoms associated with long COVID-19 may be partially explained by a neuroinflammatory process involving the activation of astrocytes and microglia [18]. The activation of microglia has been linked to adenosine triphosphate, which can act as a damage-associated molecular pattern and trigger the activation of nuclear factor (NF)-κB in microglia, leading to an upregulation of pro-inflammatory cytokines and the activation of the NLRP3 inflammasome [18,19]. The activation of the NLRP3 inflammasome results in autocatalysis and secretion of the pro-inflammatory cytokines interleukin (IL)-1β and IL-18, which may, under certain conditions, induce apoptosis in inflammatory cells [19]. Additionally, NF- κB can trigger astrogliosis, leading to neuroinflammation [18]. In addition to inflammation-driven tissue damage, severe respiratory compromise during COVID-19 may expose patients to lower blood oxygen levels and brain hypoxia, which may cause long-term damage to brain structures, including the hippocampus [20]. Therefore, pediatric patients diagnosed with severe COVID-19 that exhibit severe respiratory symptoms, such as acute shortness of breath, may be at risk of hippocampal injury and subsequent memory impairment.

In order to study the concentration deficits after COVID-19, a survey was conducted among Dutch pediatricians asking about the common long COVID symptoms noticed in their patients. It was estimated that 45% of the patients experienced concentration problems [21]. A neuropsychological test conducted on 18 patients aged 6 to 16 reported that 83% suffered from attention problems [15]. A cross-sectional study including children under 18 years diagnosed with COVID-19 found that at least 10% of the patients suffered from a lack of concentration for more than three months after the diagnosis of COVID-19 [22]. Concentration deficits can have a significant impact on a child’s academic performance. A cohort study that included a neurocognitive evaluation found that 19 out of 30 patients experienced attention impairment, 16 had executive dysfunction, and 9 reported decreased processing speed and impaired working memory [23].

Information processing speed is a critical factor in achieving high academic goals and overall social growth. Callan et al. describe that adults diagnosed with COVID-19 may experience a phenomenon called ‘brain fog’ [24]. ‘Brain fog’ is associated with adverse emotional and psychological effects and could be accountable for attention deficits and slower cognitive processing. This ‘brain fog’ is thought to result from damage to the brainstem, which may lead to breathing dysregulation and, ultimately, hypoxic brain injury [24]. Morrow et al. described in their study that four out of nine children experienced ‘brain fog’ as difficulties in focusing and prioritizing [7]. It is plausible to assume that children experience the same ‘brain fog’ phenomena as adults during long COVID syndrome.

Poor attention control is another common cognitive symptom among children with long COVID syndrome. A survey performed on 510 children aged 8 to 13 years found that 60.6% of the patients experienced a lack of concentration, almost 46% had difficulty remembering information, 40% struggled with completing everyday tasks, and 32.7% of them had processing information difficulties and experienced short-term memory problems [25]. Fatigue is another factor that can contribute to poor attention and concentration difficulties [25]. Approximately 80% of children with long COVID reported feeling fatigued, and 87% felt tired and weak [25]. Children with preexisting medical conditions, including allergies, asthma, eczema, and anxiety, have been found to report higher levels of tiredness, weakness, and fatigue. These conditions may exacerbate the impact of COVID-19 on attention control and overall well-being [25]. A cohort study of patients aged 11 to 17 found that tiredness was the most prevalent symptom 3 months after SARS-CoV-2 infection [26]. Fatigue is found to be another factor contributing to poor attention and concentration difficulties [25].

As previous studies have indicated, fatigue is a common complaint in most long COVID pediatric patients. This is not surprising, as several viruses are known to cause encephalitis in humans, which is believed to be a potential cause of chronic fatigue syndrome [27]. Myalgic encephalomyelitis/chronic fatigue syndrome (ME/CFS) is characterized by a range of symptoms, including physical tiredness, non-restorative sleep, and orthostatic intolerance, as well as cognitive impairment, which is a primary symptom of the condition [27]. The symptoms of ME/CFS overlap with those of long COVID syndrome and may potentially share the same pathophysiological mechanism. It is important to note that ME/CFS is only diagnosed when symptoms have persisted for six months or longer, whereas long COVID is expected to resolve within this time frame. Distinguishing between the two conditions is important in order to provide appropriate treatment and management [27].

### 3.2. Delirium

Delirium is an acute state of brain dysfunction characterized by mental state deterioration and altered levels of consciousness and may be triggered by certain infectious conditions [28]. According to the ‘Cornell Assessment of Pediatric Delirium’ score, approximately 34% of critical care admissions in children involve delirium [29]. A study by de Castro et al. examined the association between COVID-19 and the development of delirium [28]. Findings showed that children diagnosed with mild, medium, or severe COVID-19 experienced delirium to varying degrees, which is characterized by disorientation, confusion, and impaired recall of events. Risk factors that may increase a child’s likelihood of developing delirium include age under two years, a history of developmental delay, pre-existing comorbidities, the severity of the underlying disease, malnutrition, and mechanical ventilation [28]. One contributing factor to delirium is neuroinflammation, which is driven by a surge in cytokine levels in the central nervous system (CNS) during infection. In addition, coronavirus can directly invade into the brain and thus may cause a neuroinflammatory reaction that manifests as delirium [30]. In addition to delirium, children are at a greater risk of multiple organ failure, thrombosis, and other complications related to intensive care unit treatment [31].

A case report of two adolescents aged 16 and 17 years found that both suffered from persistent delirium after COVID-19 [31]. Both patients did not have severe respiratory symptoms or organ failure, yet they experienced delirium for at least eight days following an acute viral illness. The treatment strategy for these patients included alpha-2 agonists targeting dementia, neuroleptic agents for catatonia, and melatonin [31]. This case report suggests that neuroinflammation can directly cause brain dysfunction leading to delirium without changes in basal metabolic state. During neuroinflammation, cytokines, particularly IL-6, can pass through the blood–brain barrier, leading to CNS complications and cognitive deterioration [32]. Elevated levels of IL-4, which plays a role in memory function, may also signal neuroinflammation and result in impaired memory [32]. Although this research was conducted on adult patients, it is plausible that similar mechanisms also occur in pediatric patients, given the presence of similar patterns of long COVID syndrome in both adults and children [32].

### 3.3. Insomnia and Sleep Disturbances

Insomnia manifests with chronic symptoms of difficulty initiating sleep, waking up from sleep during the night, and early wakening, with difficulty resuming sleep easily for at least three nights a week for three months [33]. Sleep disturbances, unlike insomnia, occur in alternate intervals without a consecutive period of three months required for a diagnosis of insomnia [33]. The SARS-CoV-2 pandemic has significantly impacted the prevalence of insomnia and sleep disturbances among children. Asadi-Pooya et al. reported that out of 26 patients with long COVID syndrome, three described sleep disturbances [34].

A meta-analysis of the global assessment of sleep disturbances related to the SARS-CoV-2 pandemic found that almost 46% of children and adolescents reported sleep disturbances [35]. Along with the stressful environment created by the pandemic, long COVID syndrome brings about the emergence of persistent sleep disturbances in SARS-CoV-2 infected patients. According to Zimmermann et al., sleep disturbances occurred in 2-63% of long COVID pediatric patients [10]. A retrospective clinical case series in long COVID pediatric patients reported persistent sleep disturbances in 77.7% of the patients [15]. A study designed to identify long COVID syndrome in children found that sleep disturbances were the most prevalent symptom (18.6%) when assessed at least 60 days after SARS-CoV-2 positive diagnosis [22]. Systematic reviews on insomnia in adults have concluded that cognitive decline is more likely to occur in cases of sleep disturbances than insomnia [33], with only small to moderate deficits found in problem solving, episodic memory, and working memory observed in cases of insomnia.

### 3.4. Mental Health

Pediatric COVID-19 patients exhibit various cognitive symptoms resulting from feelings of stress and anxiety. Parents may be unaware that children can experience health-related stress and anxieties following their diagnosis of coronavirus. According to a survey by Buonsenso et al., of 510 children, 54.7% of children diagnosed with long COVID had at least three mental health issues reported by their parents [25]. These issues included difficulty in making decisions, hesitation before speaking, and difficulty completing daily tasks. It may be challenging for parents to recognize mental changes in their children. In another study by Buonsenso et al., the relationship between long COVID symptoms and the levels of distress in children was investigated [22]. The study found that most patients were not distressed by persistent symptoms, even those hospitalized due to COVID-19, but 42.6% reported distress at different levels at 120 days after COVID-19 diagnosis.

A cohort study of long COVID pediatric patients found that 23% of patients who survive mild to severe symptoms of COVID-19 exhibit mood changes [11]. In addition, the prevalence of mood changes, irritability, and anxiety or depression in children aged 5 to 9 years was significantly higher in the COVID-19 patient group than in the non-SARS-CoV-2 group. Of a sample size of 236 patients, irritability, mood changes, and anxiety or depression were found in 24.3%, 23.3%, and 13.1% of the patients, respectively. These findings suggest a remarkably high prevalence of persistent psychiatric symptoms in pediatric patients following diagnosis with coronavirus [11]. Comparison between females and males showed that females were more likely to suffer from persistent psychiatric symptoms in all three categories examined (irritability, mood changes, and anxiety or depression) [11]. The study also compared a group of pediatric patients with comorbidities, such as bronchial asthma, allergies, and congenital heart disease, to patients without comorbidities and found that irritability, mood changes, and anxiety or depression were reported more in pediatric patients with comorbidities [11].

A diagnosis of any serious illness may elicit significant anxiety in children [36]. Medical conditions such as functional abdominal pain, diabetes, asthma, and cardiovascular disorders are also associated with a higher prevalence of stress and anxiety in patients. A study found that a diagnosis of coronavirus is linked to stress and anxiety, particularly due to the uncertainty of the future of the patient’s health [36]. Korte et al. examined the concept of intolerance of uncertainty in children following a diagnosis of COVID-19. The fear of unknown consequences during a pandemic is a common concern among children, and this is correlated with the development of adverse sequelae such as stress and anxiety in particular [36].

A survey-based study involving confirmed SARS-CoV-2 pediatric patients aged 15 to 18 years was conducted to investigate the prevalence of neurocognitive deficits, pain, and mood symptoms and their association with long COVID in this patient population. The study’s results revealed that happiness was the most commonly reported mood, which is followed by tenseness and listlessness. Deficits in concentration were reported by 79.3% of the participants, and 37.8% reported experiencing fatigue. These findings suggest that the impact on mood was not directly influenced by SARS-CoV-2 infection or long COVID but rather by the lockdown measures implemented during the pandemic [37].

Gonzalez-Aumatell et al. conducted a study in which the mental health of pediatric patients was evaluated using the ‘Pediatric Symptoms Checklist’ (PSC) to identify cognitive, emotional, and behavioral problems [23]. The PSC has 35 items that can be rated as ‘never’, ‘sometimes’, or ‘often present’, which are scored 0, 1, and 2, respectively [38]. The results showed that 30% of the patients scored 30 or more points on the PSC, and 12% scored over 7 points on the attention subscale of the PSC [23]. A total of 42% of patients had a positive score on the PSC, which indicates a need for professional mental health evaluation. In a recent study, Roessler et al. examined the incidence rate of long COVID outcomes [39]. The authors found that anxiety disorder and depression had higher incidence rates in pediatric patients with long COVID compared to healthy controls. These findings are consistent with other studies that have documented persistent psychiatric symptoms in long COVID [39]. However, the authors express doubts about the mental health observations and results due to the potential for various biases influenced by temporal differences during and after the COVID-19 period. Biases may arise from differences in media coverage of COVID-19, physicians’ approach to the long COVID patients, and different social factors, all of which influence the psychological sense of danger and children’s levels of stress, making it challenging to compare across periods [39]. In light of these considerations, Graziano et al. recommend the use of psychological interventions such as cognitive reframing to help guide negative thoughts and behavioral patterns into positive ones in children [40].

## 4. Discussion

This review summarized the literature on cognitive symptoms of long COVID syndrome in children. Children may experience difficulties with attention, concentration, memory, and problem-solving abilities after contracting COVID-19. These symptoms can impact academic performance and overall quality of life. The extent and severity of these symptoms are still not fully understood. Studying and diagnosing neurocognitive symptoms in children as part of long COVID can be difficult due to the need to discern the time of initial SARS-CoV-2 infection. There are several methodological limitations in the existing studies regarding long COVID in pediatrics, such as the absence of a clear definition of long COVID, inadequate control groups, variability in follow-up times, response bias, and other forms of biases [10]. Additional difficulties may arise due to the lack of a standardized approach to diagnosing long COVID in pediatrics, inconsistent testing practices, and incorrect timing of examinations. Furthermore, the prevalence of long COVID is underestimated in children because many children may show mild or absence of symptoms during the acute phase of the infection, leading to fewer PCR tests being conducted [41]. Developing better and more unified guidelines for diagnosis and treatment is necessary.

Once a pediatric patient is diagnosed with long COVID syndrome, the treatment should be provided on multiple levels. Although a consensus on the treatment protocol has not been reached, there are recommendations for relieving the symptoms of long COVID. To address concentration deficits, memory problems, attention, and executive functions, a cognitive remediation approach is recommended using various interventions such as digital therapeutics [42,43]. Relief of anxiety, depression, and sleeping problems can be achieved with cognitive–behavioral and mindfulness-based approaches [43]. Acting on children’s thought patterns has high efficacy in shaping the way they think and behave; this includes how they perceive themselves regarding a diagnosis of coronavirus. As a result, psychologically based treatments assist in embracing positive thoughts, which is a useful tool in the prevention of stress, anxiety, and the subsequent depression and its repercussions [38]. The early detection of psychological and psychiatric symptoms has been proven to be efficient in helping patients [26,44]. Additionally, to address sleep disturbances, environmental and behavioral interventions should be considered for improving sleep hygiene [7]. Early assessment and treatment are important for delirium, which is seen more in hospitalized patients. Screening for delirium using the ABCDEFs safety bundle takes only 30 s [30] and should be considered based on the clinical picture. More information on the treatment of long COVID in pediatrics can be found in the literature on the topic [7,42,43,44].

The studies presented above show that the female sex is associated with prolonged and more severe symptoms compared to the male sex [10,11,12]. In addition, increased age is documented as a major factor for an increased risk of neurocognitive symptoms. Most of the data gathered in the studies were collected by questionnaires, which may be more reliable in older children due to their greater maturity and language skills. Comorbidities, such as asthma and congenital heart conditions, may also increase the risk of neurocognitive problems in pediatric patients [11,25,36]. These conditions may act in concert during viral illness, thereby influencing the brain and imposing additional stress and burden on individuals.

Obesity has also been identified as a risk factor for developing cognitive symptoms as part of long COVID, which is apparently because of increased expression of angiotensin converting enzyme-2 and systemic inflammatory state, leading to higher levels of pro-inflammatory cytokines such as IL-6 and tumor necrosis factor-α. These factors contribute to the development of severe and prolonged cognitive symptoms in long COVID-19 syndrome [41]. Gonzalez-Aumatell et al. reported that 66% of the pediatric patients complained of decreased school performance, while only 12% were reported to have positive findings on neurological physical examination by trained physicians [23]. This may be because subclinical impairments in concentration are more easily self-recognized than upon clinical examination. However, the difference described may also reflect an overestimation of children or parents regarding cognitive difficulties.

Finally, it is essential to use a consistent and accepted definition of long COVID in children in order to facilitate more accurate comparisons between different studies. Some of the studies included in this review identified neurocognitive symptoms as a result of long COVID syndrome but did not mention vaccination status. Further research is needed to determine the potential protective effect of vaccination on the severity of neurocognitive symptoms in long COVID syndrome. In addition, the relationship between lifestyle and long COVID syndrome has been investigated in adults [45]. It would also be valuable to examine the effect of lifestyle on neurocognitive symptoms following SARS-CoV-2 infection in children in future studies. Physical activity and diet have been shown to affect cognition and, thus, may impact the risk of developing neurocognitive symptoms during long COVID syndrome. Lastly, while it has only been reported in adults that insomnia and sleep disturbances due to long COVID may cause cognitive impairment, and the levels of the impairment may persist due to these conditions, it is important to suggest that this should also be investigated in children and adolescents.

## 5. Limitations

The findings of this review are limited by the inclusion criteria, which only allowed for studies published in the English language. Additionally, the review was based on a narrative search rather than a systematic search, which may have resulted in the omission of relevant literature. Furthermore, the search terms for post-COVID-19 syndrome were limited to ‘Post COVID-19’ and ‘long COVID-19 syndrome’, although there may be other terms used in the literature that were not included in the search.

## 6. Conclusions

In this review, we have compiled the latest reports on the potential cognitive outcomes of SARS-CoV-2 infection in children. Cognitive symptoms during the acute COVID-19 phase can persist and appear as part of long COVID syndrome. Nevertheless, the severity of such symptoms is typically mild and tolerable in most cases. There is currently limited information on the prevalence and variety of symptoms during long COVID syndrome in children and adolescents. More recent studies have found more evidence of persistent neurocognitive symptoms compared to studies conducted earlier in the pandemic. However, since some studies have a relatively low number of recruited pediatric patients due to various challenges, it is difficult to draw explicit conclusions based on them. Clinicians should be aware that the severity of COVID-19 is associated with more severe cognitive symptoms, although asymptomatic infection can also cause cognitive decline. We suggest that screening for long COVID syndrome in children should occur after recovery from the acute phase with the goal of early diagnosis and treatment to improve health outcomes.

## Data Availability

Not applicable.

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
