# Peer review of "Cognition and Mental Health in Pediatric Patients Following COVID-19"

_ijerph, 2023, doi:10.3390/ijerph20065061_

Round 1
Reviewer 1 Report (Previous Reviewer 1)
The topic of the manuscript is intriguing. However, it's more like a literature review on the topic.
Review limitations need to be added.
The last few paragraphs of the introduction should be part of the Discussion section.
There is a multitude of grammatical errors in the entire text.
Author Response
Please see the attachment.

Reviewer 2 Report (Previous Reviewer 2)
The topic is important. After modification, paper seems good and cover all aspects. Paper may be accepted now.
Author Response
Please see the attachment

Reviewer 3 Report (New Reviewer)
This article presents a very important issue. However, the article can be improved with integration of the following comments:
I suggest a minor language editing for grammatical consistency.
Line 57 - Justify the choice of 19 years as the upper borderline of inclusion.
The paper lacks discussion section. I however found that the contents of conclusion integrates a bit of discussion. I suggest that a separate discussion section should be provided. Also, the author(s) should ensure that the conclusion presents clear indications of the limitations of the study.
Author Response
Please see the attachment

This manuscript is a resubmission of an earlier submission. The following is a list of the peer review reports and author responses from that submission.
Round 1
Reviewer 1 Report
I found the topic intriguing; however, it is difficult to read because of the text's unspecific language and spelling errors.
Also, there is no mention of the age group, the test used to measure the neurological symptoms, other confounding variables etc. Therefore, it seemed incomplete.
Reviewer 2 Report
Paper had very basic information. I suggest to add some more details in terms of treatment with symptoms in children. Further, the review is missing the literature gap and how to overcome this gap.
Author may add more details about comparison in different countries and also categorization of children as well.
Reviewer 3 Report
There are major concerns related to the manuscript:
- The authors didn't use PRISMA methodology. I wonder why Moher D., Shamseer L., Clarke M., et al. Preferred reporting items for systematic review and meta-analysis protocols (PRISMA-P) 2015 statement. [Systematic Reviews Electronic Resource] 2015;4:1 http://www.prisma-statement.org/
- Line 125-131: This paragraph has nothing to do with the endpoint of study
- Refs 9 & 22 are related to respiratory long term issues. Please correct. In addition, Ref 22 is incorrect.
- Line 190-195: Ref 23 is not the primary source.
- The number of pediatric patients included in the source retrieved by Tavares-Junior is negligible. How the authors justify such statement?
PLease clarify
- This study Ref 22 Does not support author's statement: TABLE 1. Age years (range) All 59 (52–68) Mod 59 (52–67) Severe 60 (52–70)
- Line 219. Ref 24 is focused in the negative effects of the lock-down, not
in the anxiety or depression post-infection
Minor concerns: Line 233. The reference does not appear
